



**Robustness of simulating aerosol climatic impacts using regional model**
**(WRF-Chem v3.6): the sensitivity to domain size**
[1]Xiaodong Wang, [1,2,3]Chun Zhao*, [1]Mingyue Xu, [1]Qiuyan Du, [1]Jianqiu Zheng, [1]Yun Bi
[1]School of Earth and Space Sciences, University of Science and Technology of China, Hefei,
China
[2]CAS Center for Excellence in Comparative Planetology, University of Science and Technol
ogy of China, Hefei, China
[3]Frontiers Science Center for Planetary Exploration and Emerging Technologies, University
of Science and Technology of China, Hefei, China

20        *Corresponding authors:        Chun Zhao (chunzhao@ustc.edu.cn)

**Key points:**
1. Domain size has a great influence on the simulated meteorological fields and aerosol
distribution during East Asian summer monsoon (EASM).
2. Regional simulations with different domain sizes demonstrate consistently that aerosols
weaken EASM moisture transport.
3. Different domain sizes result in different strength of aerosol-induced changes of temperature
and thus circulation and rainfall over China.





**Abstract**

Domain size can have significant impacts on regional modeling results, but few studies

examining the sensitivities of regional modeling results of aerosol impacts to domain size. This
study investigates the regional modeling sensitivities of aerosol impacts on East Asian summer
monsoon (EASM) to domain size. The simulations with two different domain sizes
demonstrate consistently that aerosols induce the cooling of lower troposphere that leads to the
anti-cyclone circulation anomalies and thus the weakening of EASM moisture transport. The
aerosol-induced adjustment of monsoonal circulation results in a spatial pattern of "+-+-+" for
precipitation change over the continent of China. Domain size has a great influence on the
simulated meteorological fields. For example, the simulation with increasing domain size
produces weaker EASM circulation, which also affect aerosol distributions significantly. This
leads to the difference of simulated strength and area extent of aerosol-induced changes of
lower-tropospheric temperature and pressure, which further results in different locations of
circulation and precipitation anomalies over the continent of China. For example, over
Southeast China, aerosols induce the increase (decrease) of precipitation from the smaller-
domain (larger-domain) simulation. Different domain sizes simulate consistently aerosol-
induced increase of precipitation around 30°N over East China. This study highlights the
important impacts of domain size on regional modeling results of aerosol impacts on
circulation and precipitation, which may not be limited to East Asia. More generally, this study
also implies that proper modeling of meteorological fields with appropriate domain size is one
of the keys to simulate robust aerosol climatic impacts.









## 1. Introduction

As one of the forcing's of climate change, aerosol contributes the largest uncertainty to the total radiative forcing estimate, and it has attracted more and more attention since the 1980s (IPCC, 2013; Li et al., 2019). Aerosols can absorb and scatter solar radiation through Aerosol-Radiation interactions, affect the regional radiation budget, and amplify its impact through atmospheric mixing and circulation (e.g., Schwartz, 1996; Rinke et al., 2004; Kim et al., 2007; Z. Q. Li et al., 2010; C. Zhao et al., 2011, 2012, 2014; Myhre et al., 2013; Kuniyal et al., 2019; Zhang et al., 2020). Serving as cloud condensation nuclei or ice nuclei, aerosols can change the microscopic and macroscopic characteristics of clouds and affect the climate, which is called Aerosol-Cloud interactions (Twomey, 1977; Albrecht, 1989; Ackerman et al., 2000; Fan et al., 2012, 2013, 2016). And there are many other possible Aerosol-Cloud-Precipitation processes which may amplify or dampen this effect (Rosenfeld et al., 2008, 2014; Tao et al., 2012; Fan et al., 2015, 2018).

Due to the large population and the rapid economic development, East Asia has encountered large aerosol loading, and suffered from severe air pollution caused by various emission sources (e.g., Chan et al., 2008; X. Y. Zhang et al., 2012; Li et al., 2017; An et al., 2019). Moreover, East Asia is located in the monsoon region, the weather and climate systems are more complicated, which makes the study of aerosol effects more challenging (Ding et al., 2005; Ding, 2007; Li et al., 2016, 2019; Wu et al., 2016). In recent decades, the East Asian summer monsoon (EASM) and summer precipitation in eastern China have shown strong interdecadal changes (Ding et al., 2008, 2013; Zhou et al., 2009; Zhu et al., 2011; Zhang, 2015), which had a significant impact on agriculture, economy, and human life (An et al., 2015). Many factors are related to the interdecadal variability of the EASM, such as extraterrestrial natural forcing, internal dynamical feedbacks within the climate system and changes in atmospheric composition (e.g., greenhouse gases and aerosols) and surface conditions (land cover changes or urbanization) related to anthropogenic factors (Ding et al., 2008,2009; H. M. Li et al., 2010; Song & Zhou, 2014; Xiao & Duan, 2016; Jiang et al., 2017). As one of the forcing factors of summer climate change in East Asia, aerosol have attracted many people to study the weather and climate effects of summer aerosols in East Asia (Cowan & Cai, 2011; H. Zhang et al., 2012; Guo et al., 2013; Jiang et al., 2013, 2017; Wu et al., 2013; Song et al., 2014; Li et al., 2015, 2018; Wang et al., 2015, 2017; Chen et al., 2016; Kim et al., 2016; Xie et al., 2016; Dong et al., 2019).





Numerous studies have used global climate models to study the impacts of
anthropogenic aerosols on the EASM climate and understand the mechanisms underneath (e.g.,
Guo et al., 2013; Jiang et al., 2013, 2017; Song et al., 2014; Yan et al., 2015; Chen et al., 2016;
Wang et al., 2017; Li et al., 2018; Dong et al., 2019). The global modeling results have shown
that aerosols tend to reduce the land-sea thermal contrast, weaken the EASM, and thereby
reduce the rainfall over the continent (e.g., Guo et al., 2013; Jiang et al., 2013; Song et al., 2014;
Wang et al., 2017; Li et al., 2018; Dong et al., 2019). The reduction of monsoon precipitation
over the continent may reduce the release of latent heat from condensation in the upper
troposphere and further weaken the East Asian summer monsoon (e.g., Jiang et al., 2013; Li et
al., 2019). Jiang et al. (2013) used the CAM5 (the Community Atmospheric Model version 5)
model to study the effect of different aerosol types on East Asian summer clouds and
precipitation, and found that all anthropogenic aerosols suppressed precipitation in North China
and enhanced precipitation in South China and adjacent ocean areas. Through analyzing the
CMIP5 (Coupled Model Intercomparison Program phase 5) modeling results, Song et al. (2014)
examined the contributions of different forcings (aerosol forcing, greenhouse gas forcing,
natural forcing) to the weakening of EASM circulation during 1958–2001, and found that
aerosol forcing plays a major role in the weakening of EASM, and the contribution of natural
forcing is almost negligible, and the forcing of greenhouse gases is conducive to slightly
strengthening rather than weakening the monsoon circulation.
Global climate models have been widely used for investigating aerosol impacts,
however, there are still large uncertainties with the results at regional scale partly because the
regional-scale monsoon rainband and aerosol distributions are still not able to be described
accurately with relatively lower model horizontal resolution (H. M. Li et al., 2010; Guo et al.,
2013; Jiang et al., 2013; Song et al., 2014; Li et al., 2018; Dong et al., 2019). In comparison,
regional model often has higher horizontal resolution and can better capture regional features
of weather and climate systems and aerosol distributions, and therefore has been used to
investigate aerosol climatic impacts recently (e.g., Zhao et al., 2011, 2012; Wu et al., 2013;
Wang et al., 2015; Zhuang et al., 2018). For example, using the regional model (RegCCMS),
Wang et al. (2015) found that aerosol-cloud interaction decreases the autoconversion rates of
cloud water to rain water and increases the liquid water path of clouds in East China,
strengthens the cooling of lower atmosphere caused by the direct radiation effect, and
suppresses the convective precipitation. Wu et al. (2013), with the regional model (WRF-
Chem), found that the aerosol heating effect caused the cloud to move northward over East
China and led to the increased precipitation in the north.





Although regional model at higher horizontal resolution may better capture regional
features of wind, cloud, precipitation, and aerosol, it also introduces additional uncertainties
on modeling regional aerosol climatic impacts resulted from the lateral boundary of regional
simulation. Previous studies have found that domain size of regional model can significantly
influence the simulation results (e.g., Warner et al., 1997; Leduc and Laprise, 2009; Leduc et
al., 2011; Bhaskaran et al., 2012; Giorgi, 2019). For example, Bhaskaran et al. (2012) studied
the sensitivity of the simulated hydrological cycle to the regional domain size over the Indian
subcontinent. They found that the simulations with smaller domains produced increased
precipitation and evapotranspiration on seasonal mean and higher number of moderate
precipitation days relative to the ones with larger domains. Different distributions of cloud,
precipitation, and winds from the simulations with different domain sizes may lead to different
aerosol distributions and its climatic impacts. Previous studies have found that aerosol impacts
on precipitation, clouds, and circulation will be significantly different under different weather
and climate conditions (e.g., Wu et al., 2013; Wang et al., 2015; Xie et al., 2016). In addition,
Seth and Giorgi. (1998) found that the smaller-domain simulation produced better precipitation
compared with the observations, but resulted in an unrealistic response to the internal forcing.
This indicates that the simulation domain size may also affect the aerosol impacts on large-
scale circulation. Therefore, the regional simulation with increased domain size may be
preferred to better reflect the overall aerosol impacts on large-scale circulation and weather
system without the strict constraint from the boundary forcing (e.g., Seth and Giorgi, 1998;
Leduc and Laprise, 2009; Xue et al., 2014), but the increased domain size may make the
simulations deviated from the forcing such as the reanalysis.
As far as we know, there are few studies examining the sensitivities of regional
modeling results of aerosol impacts to regional domain size. Although it can be expected that
domain size will play a role, it is not known to what extent and how domain size can affect
modeling results of aerosol climatic impacts. Therefore, in this study, the regional online-
coupled meteorology and chemistry model WRF-Chem (Weather Research and Forecasting
model coupled with Chemistry) (Grell et al., 2005; Skamarock et al., 2008) is used to study the
aerosol impacts on East Asian summer monsoon system and focus on the modeling sensitivities
to regional domain size. WRF-Chem has been widely used for studying aerosol meteorological
and climatic impacts over East Asia (e.g.,A. J. Ding et al., 2013; Wu et al., 2013; Gao et al.,
2014; Chen et al., 2014; Zhao et al., 2014; Huang et al., 2016; Liu et al., 2016; Petaja et al.,
2016; Zhao B et al., 2017). The investigation of aerosol impacts under different simulated
meteorological fields due to different domain sizes may also help understand the different



modeling results about the aerosol impacts on East Asian summer monsoon from previous
studies. The study is organized as follows. Section 2 describes the numerical experiments and
methods. The results and discussions are presented in Section 3. A summary is provided in
Section 4.

**2. Methodology**
**2.1 WRF-Chem**
In this study, the version of WRF-Chem updated by the University of Science and
Technology of China (USTC version of WRF-Chem) is used. The model simulates the
emission, transport, mixing, and chemical transformation of trace gases and aerosols
simultaneously with the meteorology, and can be used for investigation of regional-scale air
quality and interactions between meteorology and chemistry. Compared with the publicly
released version, the USTC version of WRF-Chem includes a few additional functions, such
as the diagnosis of radiative forcing of aerosol species, optimized Kain-Fritsch (KF) convection
scheme, aerosol-snow interaction, land surface coupled biogenic Volatile Organic Compound
(VOC) emission, etc. (Zhao et al., 2013a, b, 2014, 2016; Hu et al., 2019; Du et al., 2020), all
of which may have important impact on modeling aerosol and its climatic impacts.
The Model for Simulating Aerosol Interactions and Chemistry (MOSAIC) aerosol
module coupled with CBM-Z (carbon bond mechanism) photochemical mechanism in WRF-
Chem is selected in this study (Zaveri & Peters, 1999; Zaveri et al., 2008). MOSAIC uses a
sectional approach to represent aerosol size distributions with four or eight discrete size bins
in the current version of WRF-Chem (Fast et al., 2006). To reduce the computational cost, four
discrete size bins is selected in this study. All major aerosol components including sulfate,
nitrate, ammonium, black carbon, organic matter, sea-salt, mineral dust, and other inorganic
matter (OIN) are simulated in the model. The MOSAIC aerosol scheme includes physical and
chemical processes of nucleation, condensation, coagulation, aqueous-phase chemistry, and
water uptake by aerosols. Dry deposition of aerosol mass and number is simulated following
the approach of Binkowski and Shankar (1995), which includes both turbulent diffusion and
gravitational settling. Wet removal of aerosols by grid-resolved stratiform clouds and
precipitation includes in-cloud removal (rainout) and below-cloud removal (washout) by
impaction and interception, following Easter et al. (2004) and Chapman et al. (2009). In this
study, cloud-ice-borne aerosols are not explicitly treated in the model, but the removal of
aerosols by the droplet freezing process is considered. Convective transport and wet removal
of aerosols by cumulus clouds is coupled with the Kain-Fritsch cumulus scheme as Zhao et al.
(2013b). Aerosol radiative feedback is coupled with the Rapid Radiative Transfer Model
(RRTMG) (Mlawer et al., 1997; Iacono et al., 2000) for both SW and LW radiation as
implemented by Zhao et al. (2011). The optical properties and direct radiative forcing of
individual aerosol species in the atmosphere are diagnosed following the methodology
described in Zhao et al. (2013a).

**2.2 Numerical experiments**

Four sets of experiments CTRL-L, CTRL-S, CLEAN-L, CLEAN-S with different
simulation domain sizes or emission configurations are conducted as listed in Table 1. The
control (CTRL-S, CTRL-L) simulations use standard anthropogenic emission dataset
(described in Section 2.3), while the clean simulations (CLEAN-S, CLEAN-L) apply a factor
of 0.1 on the standard emissions within the small domain to represent a clean atmosphere
condition over East Asia (Fig. 1). The CTRL-L and CTRL-S (CLEAN-L and CLEAN-S)
represent the simulations with large and small domain sizes, respectively, as shown in Figure
1. The aerosol impacts can be calculated by the difference between the CTRL and CLEAN
simulations for each simulation domain. The comparison of aerosol impacts between the large
and small simulation domains implies the sensitivity of aerosol impacts to domain size.
All the WRF-Chem experiments select the Morrison two-moment microphysics
(Morrison et al., 2009), Kain-Fritsch cumulus scheme (Kain, 2004), unified Noah land-surface
model, Rapid Radiative Transfer Model (RRTMG) longwave and shortwave radiation schemes
(Iacono et al., 2008), and MYNN planetary boundary layer (PBL) scheme (Nakanishi & Niino,
2006,2009). Following Du et al. (2020), the PBL mixing coefficient is modified to simulate
better PBL mixing of aerosols. Five ensemble simulations are performed for each experiment
by changing the initial conditions at UTC 0000 from May 12 to May 16, 2017. The averaged
results from five ensembles are analyzed to reduce the influence of modeling internal
variability. The simulations run through entire June and July of 2017. The analysis focuses on
the simulation results for June 1 to July 31, 2017. The meteorological initial and lateral
boundary conditions are derived from National Centers for Environmental Prediction (NCEP)
Final (FNL) Operational Global Analysis data (NCEP, 2000) with a resolution of 1°×1° and a
time resolution of 6h. The chemical initial and boundary conditions are provided by a quasi-
global WRF-Chem simulation for the same time period. The quasi-global WRF-Chem
simulation is performed at 1°×1° horizontal resolution with 360×130 grid cells (180°W-
180°E, 60°S-70°N). More details about the general configuration of a quasi-global WRF-Chem





simulation can be found in Zhao et al. (2013b) and Hu et al. (2016). The simulation
configuration is summarized in Table 2.
**2.3 Emissions**
Biomass burning emissions are obtained from the Fire Inventory (FINN) of the National
Center for Atmospheric Research with hourly temporal resolution and 1 km horizontal
resolution (Wiedinmyer et al., 2011), and the injection heights follow Dentener et al. (2006)
for the Aerosol Comparison between Observations and Models (AeroCom) project. The natural
dust emission fluxes are calculated based on the adjusted GOCART dust emission scheme
(Ginoux et al., 2001; Zhao et al., 2010), and the emitted dust particles are distributed into the
MOSAIC aerosol size bins following a theoretical expression based on the physics of scale-
invariant fragmentation of brittle materials derived by Kok (2011). More details about the dust
emission scheme coupled with MOSAIC aerosol scheme in WRF-Chem can be found in Zhao
et al. (2010, 2013b). Sea-salt emission follows Zhao et al. (2013a), which includes correction
of particles with radius less than 0.2 μm and dependence of sea-salt emission on sea surface
temperature. Anthropogenic emissions are obtained from the Multi-resolution Emission
Inventory for China (MEIC) at 0.1°x0.1° horizontal resolution and with monthly temporal
resolution for 2015 (Li et al., 2017; Zheng et al., 2018), except that the emissions outside of
China are from the Hemispheric Transport of Air Pollution version2 (HTAPv2) at 0.1°x0.1°
horizontal resolution and with monthly temporal resolution for the year 2010 (Janssens-
Maenhout et al., 2015) (Fig. 1). As discussed above, the anthropogenic emissions in the
CLEAN experiments is a factor of 0.1 of that in the CTRL experiment, and in the CLEAN-L
experiment, only the emissions in the area of the small domain (denoted by the red box) are
adjusted. In this way, the emission reduction from the simulations with both domains are made
consistent.

**3. Results**
**3.1 Sensitivity of simulated meteorological fields to domain size**
Figure 2 shows the spatial distributions of precipitation and moisture transport at 700
hPa over the small domain averaged for June and July of 2017 from the observation and
reanalysis, and the simulations of CLEAN-S and CLEAN-L. The observation and reanalysis
show that the southwesterly transports large amount of moisture into East China. The converge
of large amount of moisture results in heavy precipitation over southern China and its adjacent
ocean. Due to the gradual weakening of northeastward moisture transport and the blocking



effect of the western mountains, precipitation becomes much weaker over northern and western
China. Compared with the CMORPH observation and ERA5 reanalysis (Fig. 2), CLEAN-S
can reasonably produce the spatial distribution of precipitation and moisture transport at 700
hPa, with slight underestimation of meridional moisture transport over eastern China. It is
evident that the meridional moisture transport over southern China becomes weaker with the
increasing domain size, and the eastward transport becomes stronger. In addition, the overall
southwesterly moisture transport shift to the east. This leads to a decrease of precipitation over
eastern China and an increase over the East China Sea. Compared with the observations of
hourly precipitation from the CMA stations over eastern China (Fig. S1 in the supporting
material), both the CLEAN-S and CLEAN-L experiments can generally reproduce the daily
variation of precipitation over eastern China, although the CLEAN-L simulated precipitation
is lower consistent with its weaker moisture transport over the region.
The difference in moisture transport between the simulations with different domain
sizes results from their difference in geopotential height and wind circulation. Figure 3 shows
the spatial distributions of geopotential height (GPH) and wind field at 700 hPa from the
CLEAN-S simulation, and the difference between CLEAN-L and CLEAN-S. The comparison
with the ERA5 reanalysis shows that the CLEAN-S can well simulate the distributions of GPH
and wind fields at 700 hPa (Fig. S2 in the supporting material). The spatial distribution of wind
fields is generally consistent with that of moisture transport (Fig. 2), and is largely controlled
by the West Pacific sub-tropical high (WPSH). Compared to CLEAN-S, CLEAN-L simulates
lower GPH at 700 hPa and produces an anomalous lower pressure center on the East China
Sea, which indicates the weaker WPSH with increasing domain size. This causes the
southwestward wind anomalies over the continent, which weakens the monsoon driven
northeastward moisture transport. Over the South China Sea, the westerly anomalies enhance
the eastward transport of moisture.
The impact of domain size is not only on the horizontal distribution of wind fields but
also on the vertical circulation. Figure 4 shows the cross-section of meridional temperature
anomalies and wind averaged for 105°E and 122°E from the CLEAN-S simulation, and the
difference of temperature (not meridional temperature anomalies) and wind between CLEAN-
L and CLEAN-S. The meridional temperature anomalies are calculated by subtracting the
mean temperature in this latitude range (as shown in Fig. 4) at each pressure level. First of all,
CLEAN-S can general reproduce the temperature gradient and wind circulation from the ERA5
reanalysis (Fig. S3 in the supporting material). Relatively large meridional temperature
gradient exists between 700 hPa and 200 hPa, where the temperature is higher over the South.



Below 700 hPa, the temperature gradient is relatively weaker, and the temperature is higher
over the North. Along with this distribution of temperature gradient, meridional wind blows
from the South and the North and converges at the latitude around 34°N, which generates
strong upward motion in the area of 20°N-35°N. This is consistent with the spatial distribution
of precipitation and moisture transport (Fig. 2). Compared with the CLEAN-S experiment, the
CLEAN-L experiment produces larger meridional temperature gradient between 700 hPa and
200 hPa and weaker gradient below 850 hPa. The circulation from the CLEAN-L is generally
consistent with CLEAN-S, but the southerly wind from CLEAN-L is weaker and the northerly
wind is stronger. This results in an overall northerly wind anomalies from CLEAN-L compared
with CLEAN-S, and also a southward shift of the wind convergence from 34°N to 32°N. It is
also noteworthy that the upward motion is weakened around 22°N-38°N and strengthened to
the south of 20°N due to the increased domain size.

**3.2 Sensitivity of simulated aerosol characteristics to domain size**
Figure 5 shows the spatial distribution of AOD averaged for June and July of 2017 from
the CTRL-S simulation, and the difference between CTRL-L and CTRL-S. It can be seen that
relatively high AOD (>0.6) exists in the Sichuan Basin and the North China Plain. The AOD
over East Central China and South China is relatively lower (0.2-0.5), which is in line with
previous research (Luo et al., 2014; Qi et al., 2013). In general, the CTRL-S generally captures
the spatial distribution of retrieved AOD from MISR (Fig. S4 in the supporting material).
Compared with the CTRL-S experiment, CTRL-L simulates a similar spatial pattern of AOD
as CTRL-S, but produces higher AOD in southern China and lower AOD in most areas of
northern China. To explore the reasons of difference between the two simulations, Figure 6
shows the spatial distributions of column integrated total PM2.5 concentration and water
content in aerosol averaged for June and July of 2017 from the CTRL-S simulation, and the
difference between CTRL-L and CTRL-S. The CTRL-S simulation shows high PM2.5 mass
loading over North China Plain, which is consistent with the spatial distribution of AOD (Fig.
5). The PM2.5 mass loading also shows high values over Northwest China, which is not shown
in the spatial distribution of AOD. This is mainly due to the high mass loading of dust over
Northwest China (Fig. S5 in the supporting material), and the water content associated with
dust is relatively small.
CTRL-L simulates higher PM2.5 mass loading over Southeast China and lower values
over North China, which is consistent with AOD. The difference of water content in aerosol
shows similar pattern. The analysis shows that the difference of PM2.5 mass loading over



North China is mainly due to the difference of dust, while the difference over Southeast China
is due to anthropogenic aerosols (Fig. S5). The reduction of dust mass loading over North
China from CTRL-L is primarily due to its weakening of westerlies over Northwest China
compared to CTRL-S (Fig. 3), which results in less transport of dust into the downwind region.
The increase of aerosol mass loading over Southeast China in CTRL-L is partly due to its less
wet scavenging associated with weak precipitation (Fig. 2). The weakening of northward
transport of aerosol (Fig. 3) also contributes to the increase of PM2.5 mass loading over
southern China in CTRL-L. Besides the change of dry aerosol mass loading, the change of
water content in aerosol between the two experiments also contributes to the change in AOD,
which results from the difference of both dry aerosol mass and moisture.

Figure 7 shows the latitude-height cross-section of total PM2.5 averaged between

105°E and 122°E for June and July of 2017 from the CTRL-S experiment, and the difference
between CTRL-L and CTRL-S. The latitudinal distribution of aerosol is consistent with it
spatial pattern with high aerosol mass concentration over North China. The mass concentration
gradually reduces from the surface to the free atmosphere. The mass concentration around 500
hPa over North China can reach 5 ug/m$^3$ that is comparable to the surface concentration over
South China. In general, CTRL-L simulates higher aerosol mass concentration over South
China and lower aerosol mass concentration over North China from the surface to about 500
hPa. At 32°N-36°N, CTRL-L simulates lower aerosol mass concentration near the surface and
higher above 850 hPa. The difference of aerosol horizontal and vertical distributions and also
the circulation patterns between the two experiments may lead to the difference in simulating
aerosol impacts on East Asian monsoon system.

**3.3 Sensitivity of aerosol impact to domain size**

Before studying the sensitivity of aerosol impacts to domain size, the impacts of aerosol

on precipitation and circulation from the small domain simulations are first investigated. Figure
8 shows the spatial distributions of aerosol-induced difference (CTRL-CLEAN) of
precipitation and moisture transport at 700 hPa averaged for June and July of 2017 from the
small domain simulations. The dominant effect is that aerosol weakens the southwesterlies
flow and reduces the moisture transport over the continent of Central and South China
(primarily between 105°E-115°E). Along the coast of Southeast China, the moisture transport
is enhanced slightly. Over the continent of China, aerosol induces a "+-+-+" pattern of
precipitation changes, i.e., precipitation increases in the south of 25°N, north of 40°N, and
around 30°N, while decreases at 25°N~30°N and 32°N~40°N. This weakening of monsoonal



circulation at the lower troposphere is found mainly due to the cooling of lower troposphere
and thus the increase of surface pressure by aerosols (Fig. 9). The temperature averaged for
lower troposphere (below 500 hPa) is reduced by aerosols over the continent of China, which
results in a positive pressure anomaly center in Southwest China. This leads to an anticyclone
anomaly as shown in Fig. 8, which weakens the monsoonal southwesterlies between 105°E-
115°E.

In order to further understand the mechanisms of aerosol impacts and isolate aerosol-

radiation and aerosol-cloud interactions, another set of numerical experiment (NoRA-S) with
the small domain are conducted, similar as CTRL-S but with the aerosol-radiation interaction
turned off. The difference of results between NoRA-S and CLEAN-S (NoRA-S minus
CLEAN-S) is interpreted as the impacts of aerosol-cloud interaction, while the difference of
results between CTRL-S and NoRA-S (CTRL-S minus NoRA-S) is interpreted as the impacts
of aerosol-radiation interactions. Figure 10 shows the spatial distribution the impacts of
aerosol-cloud and aerosol-radiation interactions on (a, d) tropospheric temperature averaged
below 500 hPa, (b, e) surface pressure, (c, f) precipitation and moisture transport. The aerosol-
cloud interaction reduces significantly the lower tropospheric temperature (Fig. 10a) over a
large area of South China (to the south of 32°N) due to its increasing of cloud amounts (Fig.
S6a in the supporting material) over this area, which results in an increase of surface pressure
in this area (Fig. 10b). Similarly, aerosol-cloud interaction also increases cloud amounts over
Northeast China and its adjacent ocean (Fig. S6a) and thus reduces the lower tropospheric
temperature and increases the surface pressure over the area.  The surface pressure over the
Yellow River Basin is reduced slightly by aerosol-cloud interaction due to the reduction of
cloud amounts (Fig. S6a) and the increase of lower tropospheric temperature. The difference
between NoRA-S and CLEAN-S over Northwest China is due to the dust-radiation interaction
that is included in CLEAN-S but not in NoRA-S. The analysis of this study focuses on the
impacts of anthropogenic aerosol. The combined effect of two anti-cyclone anomalies due to
the two positive pressure anomalies at the lower troposphere results in the southward wind
anomalies over the ocean and the northward wind anomalies over North China, while the
changes of circulations in other areas of China is negligible.

The primary impacts of aerosol-radiation interaction on lower-atmospheric temperature

are the positive temperature anomalies over the Yellow Ocean and over central China and the
negative temperature anomalies over the Yellow River Basin and Southwest China, which is
the combined effects from the aerosol cooling at the surface and heating in the atmosphere and
also the adjustment of cloud distributions (Fig. S6b and Fig. S7). The two positive temperature



anomaly centers lead to two negative pressure anomaly centers and thus a large cyclone
circulation anomaly over the continent of East China. Therefore, it can be noted that the
influence of aerosol-cloud and aerosol-radiation interactions on monsoonal circulations are
counteracted over the ocean and over northern China, which results in relatively small changes
of monsoonal circulation over the ocean and over northern China (Fig. 8). The overall aerosol
impact is shown as the weakening of the monsoonal circulation over the continent of central
and southern China (Fig. 8), which is mainly contributed by the aerosol-radiation interaction.

Figure 11 shows the latitude-pressure cross-section of aerosol-induced difference

(CTRL-CLEAN) of temperature and wind averaged between 105°E and 122°E for June and
July of 2017 from the small domain simulation. It can be seen that the pattern of precipitation
change corresponds well to the changes of wind circulation. The weakening of monsoonal
southwesterlies result in a sinking airflow anomaly around 28°N and the compensating upward
anomaly around 24°N in the south of China, and also a downdraft around 35°N and an updraft
around 40°N in north China. These two sinking airflows corresponds to the reduced
precipitation between 25°N and 30°N and between 32°N and 40°N, respectively (Fig. 8), while
these updrafts correspond to the increasing precipitation between 22°N and 25°N and between
32°N and 40°N. There is also weak upward compensating airflow around 30°N, leading to the
slight increase of precipitation in the area (Fig. 8). It is noteworthy that aerosols lead to an
abnormal cooling center around 33°N between 400 hPa to 200 hPa. This is mainly because of
less solar radiation entering the atmosphere due to aerosol-radiation and aerosol-cloud
interactions, and also weaker monsoonal airflow that leads to less release of latent heat from
cloud and precipitation (Fig. S8 in the supporting material). This cooling anomaly center also
strengthens the downdraft anomalies on its both sides, further weakening the monsoonal
circulation.

In order to explore the sensitivity of aerosol impacts to domain size, similar as Fig. 8,

Figure 12 shows the results from the large domain simulations. One consistent signal between
the simulations with different domain sizes is that aerosols weaken the southwesterlies flow
and reduce the moisture transport over the continent of Central and South China. The difference
is that this weakening is not only over the inland of China but also extending to over the South
China Ocean. The weakening of monsoon airflow is broader with the increasing domain size,
which may be due to its weaker monsoon airflow (Fig. 3) and less constraint from the lateral
boundaries in the large domain simulation. Another consistent signal between the two sets of
simulations with different domain sizes is that aerosol induces a similar "+-+-+" pattern of
precipitation changes over the domain, except that the areas with precipitation reduction



432 become broader. This leads to the precipitation reduction over almost the entire region between

433 20°N~40°N over the continent of China except the area around 30°N with increasing

434 precipitation. The increases of precipitation on the two sides of precipitation reduction area

435 shift southward to the South China ocean and northward to the north of 40°N, respectively.

436   Similar as the small domain simulation, the weakening of monsoonal airflow in the

437 large domain simulation is also due to the abnormal positive lower-level pressure that is caused

438 by the lower atmosphere cooling (Fig. 13), which can also be explained by the effects of

439 aerosol-radiation and aerosol-cloud interactions (Fig. S9 and Fig. S10 in the supporting

440 material). However, compared with the small domain simulation (Fig. 9), the cooling anomaly

441 of lower-tropospheric temperature and thus the positive anomaly of lower-level pressure cover

442 a broader area from the large domain simulation. The two aerosol-induced cooling centers over

443 the continent of China lead to two positive lower-level pressure anomaly that results in a large

444 anti-cycle circulation anomaly (Fig. 12), which weakens the monsoonal southwesterly airflow

445 over South China and the South China Ocean and also slightly enhances the southwesterly over

446 West China. Again, the pattern of precipitation change corresponds well to the changes of wind

447 circulation (Fig. 14). With larger domain size, aerosols lead to a broader area (between

448 20°N~40°N) of abnormal cooling in the troposphere up to 200 hPa. The single cooling center

449 in the small domain simulation is split into two centers, one around 30°N at 250 hPa and

450 another around 36°N at 700 hPa. The weakening of the background circulation and broader

451 cooling area lead to the broader sinking airflows over the region, which results in the broader

452 area of reduced precipitation compared with the small domain simulation (Fig. 8 and Fig. 12).

453 The increasing precipitation around 30°N is also resulted from the compensating updrafts

454 around 30°N.

## 456 4. Summary and Discussion

457   Due to the importance of domain size on regional modeling results and few studies

458 examining the sensitivities of regional modeling results of aerosol impacts to domain size, this

459 study applies the WRF-Chem model to simulate the anthropogenic aerosol impacts on East

460 Asian summer monsoon circulation and precipitation, focusing on the modeling sensitivities to

461 regional domain size. The impacts of domain size on meteorological fields, aerosol

462 characteristics, and aerosol impacts are investigated.

463   First of all, the domain size has a great influence on the simulated meteorological fields.

464 From the small domain simulation, the circulation and precipitation are in good agreement with



the reanalysis data and observations. The large domain simulation produces weaker East Asian
summer monsoon system shifting southward, which results in the precipitation decrease in
southern China and increase in the adjacent ocean. The changes of circulation and precipitation
also lead to the increase of aerosol mass loading in southern China and the decrease in northern
China in the large domain simulation. The deviation of atmospheric fields particularly the
circulation between the simulations with different domains is partly due to their different
constraint from lateral boundary conditions. With less constraint of the boundary forcing from
the reanalysis data, the large domain simulation may produce negative bias in precipitation
over the Yangtze River Basin and positive bias in water vapor transport over the South China
Ocean as previous studies. The uncertainties in moisture transport prescribed in the lateral
boundaries from the reanalysis over a larger domain may also contribute to the biases (e.g.,
Wang and Yang, 2008; Huang and Gao, 2018). With the larger domain, the simulation includes
larger area of ocean. Without considering the online interaction between the atmosphere and
the ocean (i.e., with prescribed SST from the reanalysis), the artificial positive feedback
between precipitation and surface latent heat flux may overestimate the precipitation over the
subtropical Western North Pacific (WNP) and inhibit the westward expansion of the WNP
subtropical high (e.g., Cha and Lee, 2009; Lee and Cha, 2020).
In terms of the climatic impacts of anthropogenic aerosols on East Asian summer
monsoon, as shown in the schematic plot (Fig. 15), aerosols induce the cooling of lower
troposphere over the continent through aerosol-radiation and aerosol-cloud interactions, which
leads to an increase of regional pressure at lower atmosphere. The regional positive pressure
anomalies result in the anti-cyclone circulation anomalies and thus weakens the summer
monsoonal northeastward moisture transport, which is consistent with previous studies (e.g.,
Y. Q. Jiang et al., 2013; Song et al., 2014; T. J. Wang et al., 2015; Xie et al., 2016). The
weakening of monsoonal circulation leads to several sinking airflows and compensating
updrafts that correspond well to the regions with the decrease and increase of precipitation,
respectively, showing a spatial pattern of "+-+-+" for precipitation change. The difference in
the aerosol impacts from the numerical experiments with different domain sizes is mainly
determined by their simulated different strength and area extent of the aerosol-induced lower-
tropospheric negative temperature anomalies. Compared with the smaller-domain simulation,
the larger-domain simulation with weaker monsoonal circulation generates a broader area with
negative temperature and positive pressure anomalies at the lower troposphere, which results
in broader sinking airflows and thus broader areas of precipitation reduction over the continent
of China. This could lead to the opposite signals of precipitation change due to aerosols over



China, for example, over Southeast China, the increase and decrease of precipitation from the
smaller-domain and larger-domain simulations, respectively. The consistent signal of aerosol
impacts between the simulations with different domain sizes is the increasing precipitation
around 30°N that is resulted from the compensating updraft over the region.

Although the modeling results of aerosol impacts in this study may have some

uncertainties associated with physical and chemical processes, emissions, and simulation
horizontal resolutions, it highlights the impacts of simulation domain size on regional modeling
aerosol impacts on monsoonal circulation and precipitation, which may not be limited to the
region of East Asia. Uncertainties of modeling aerosol climatic impacts are often investigated
focusing on aerosol characteristics such as their distributions and properties. This study adds
another complexity (impact of domain size) on regional modeling of aerosol climatic impacts.
More specifically, although larger-domain simulation may better allow feedbacks of aerosol
impacts on weather and climate systems without strong lateral boundary constraint (e.g., Seth
and Giorgi, 1998; Leduc and Laprise, 2009), it may produce biased meteorological fields
compared to smaller-domain simulation, which can significantly influence the modeling results
of aerosol impacts. It may be the key to simulate reasonable/less biased meteorological fields
with larger regional domain or global domain in order to model robust aerosol climatic impacts.
More generally, this study also highlights the impacts of background meteorological fields
(without aerosol effect) on simulated aerosol impacts. Proper modeling of background
meteorological fields is one of the keys to simulate robust aerosol climatic impacts. The model
inter-comparison study of aerosol climatic impacts should also focus on the diversity of
simulated background meteorological fields besides aerosol characteristics.


### 523    Code and data availability

The release version of WRF-Chem can be downloaded from

http://www2.mmm.ucar.edu/wrf/users/download/get_source.html. The code of updated
USTC version of WRF-Chem is available at https://doi.org/10.5281/zenodo.4663508 or
contact chunzhao@ustc.edu.cn. The dataset from the European Centre for Medium-Range
Weather Forecasts (ECMWF) Reanalysis (ERA5) can be downloaded from
https://rda.ucar.edu/datasets/ds633.1/ (last access: Aug 2021). The CMORPH data can be
downloaded from https://ftp.cpc.ncep.noaa.gov/precip/CMORPH_V1.0/CRT/0.25deg-
DLY_00Z/2017/ (last access: Aug 2021).






## Author contributions

Xiaodong Wang and Chun Zhao designed the experiments, conducted and analyzed the
simulations. All authors contributed to the discussion and final version of the paper.

## Acknowledgements

This research was supported by the National Natural Science Foundation of China
(42061134009, 41775146, 91837310), the USTC Research Funds of the Double First-Class
Initiative (YD2080002007), Fundamental Research Funds for the Central Universities
(WK2080000101), and the Strategic Priority Research Program of Chinese Academy of
Sciences (XDB41000000). The study used the computing resources from the High-
Performance Computing Center of University of Science and Technology of China (USTC)
and the TH-2 of National Supercomputer Center in Guangzhou (NSCC-GZ).






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

0767.1












**Table 1.** Experiment Description.

| Experiment ID | Experiment Description |
|---|---|
| CTRL-L | Control experiment with large simulation domain. |
| CLEAN-L | Same as CTRL-L, but the anthropogenic aerosol emissions are 0.1 times of CTRL-L. |
| CTRL-S | Control experiment with small simulation domain. |
| CLEAN-S | Same as CTRL-S, but the anthropogenic aerosol emissions are 0.1 times of CTRL-S. |






**Table 2.** Summary of model configurations.

| Description | Selection(L, S) |
|---|---|
| Horizontal grid spacing | 30km |
| Grid dimensions | 201x231, 121x121 |
| Vertical layers | 41 |
| Topography | USGS_30s |
| Model top press | 100hPa |
| Aerosol scheme | MOSAIC 4 bin |
| Gas-phase chemistry | CBM-Z |
| Long wave Radiation | RRTMG |
| Short wave Radiation | RRTMG |
| Cloud Microphysics | Morrison 2-moment |
| Cumulus Cloud | Kain-Fritsch |
| Planetary boundary layer | MYNN 3rd |
| Land surface | unified Noah land-surface model |
| Meteorological Forcing | FNL, 1°x1° ,6 hourly |










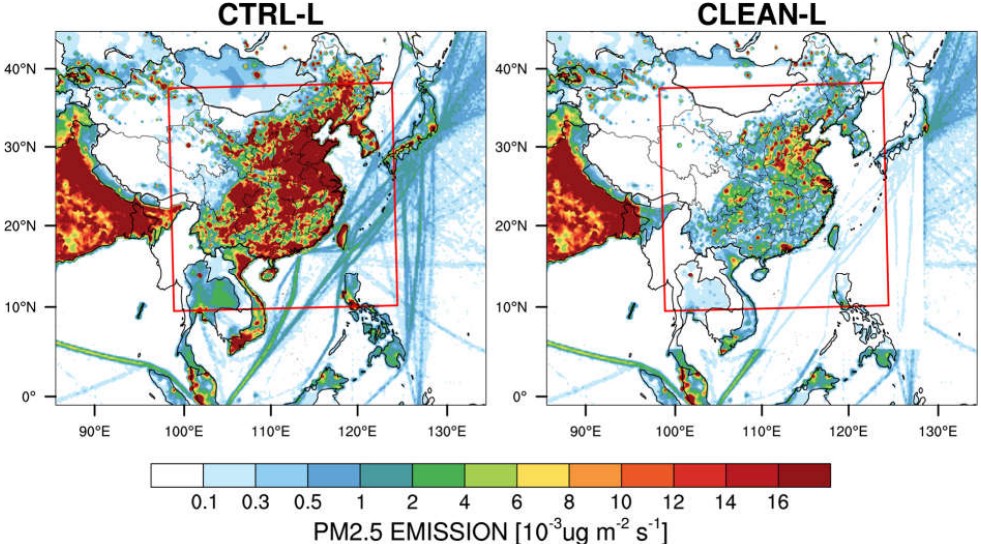


**Figure 1.** Spatial distributions of anthropogenic emissions of primary PM2.5 averaged for June and July for the simulation domains. The red box in the large simulation domain represents the small domain.





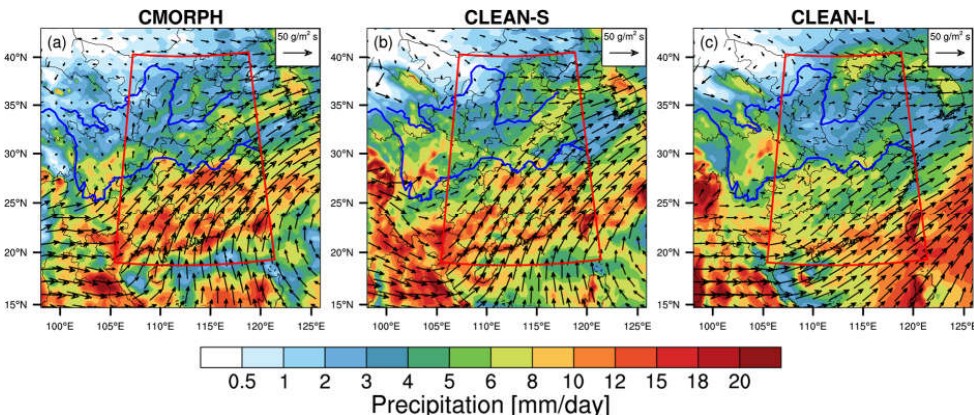

**Figure 2.** Mean precipitation rate (mm/day) and 700hPa moisture transport (g/m² s) over the
small domain for the two months of June and July 2017 from (a) CMORPH and ERA5
reanalysis, (b) CLEAN-S simulation, and (c) CLEAN-L simulation. The red box (20°N-42°N,
105°E-122°E) represents the focus area of analysis in follow. (a) Precipitation data comes from
CMORPH, and the 700hPa moisture transport field data is obtained by processing ERA5
reanalysis.




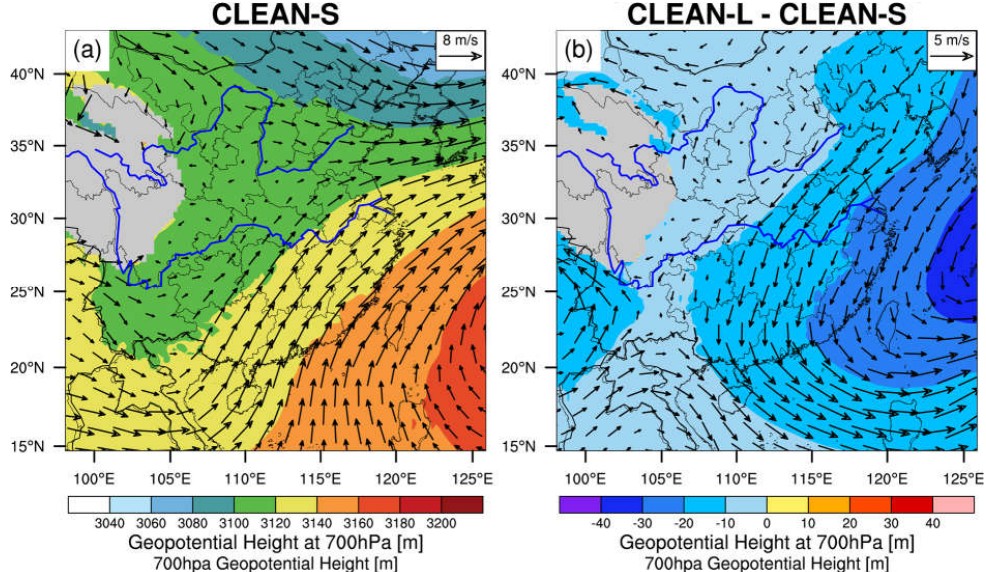

**Figure 3.** Spatial distribution of mean 700 hPa Geopotential Height and winds of June and July 2017 from (a) CLEAN-S, and the (b) difference between CLEAN-L and CLEAN-S.




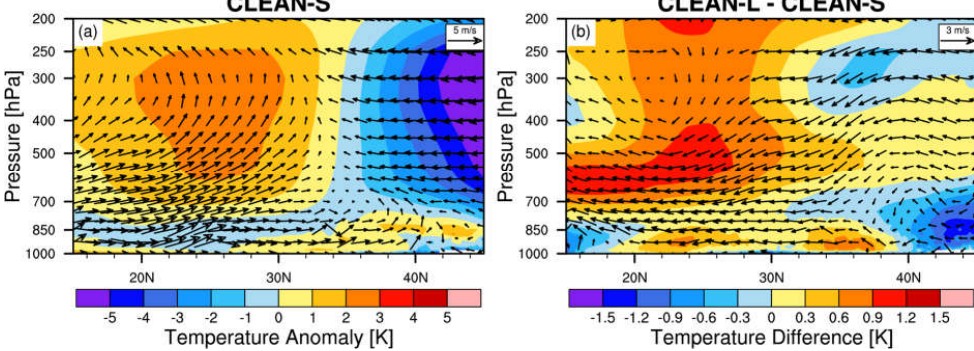

**Figure 4.** The cross-section of meridional temperature anomalies and wind averaged for 105°E and 122°E from (a) the CLEAN-S simulation, and (b) the difference of temperature (not meridional temperature anomalies) between CLEAN-L and CLEAN-S. The meridional temperature anomalies are calculated by subtracting the mean temperature in this latitude range at each pressure level.






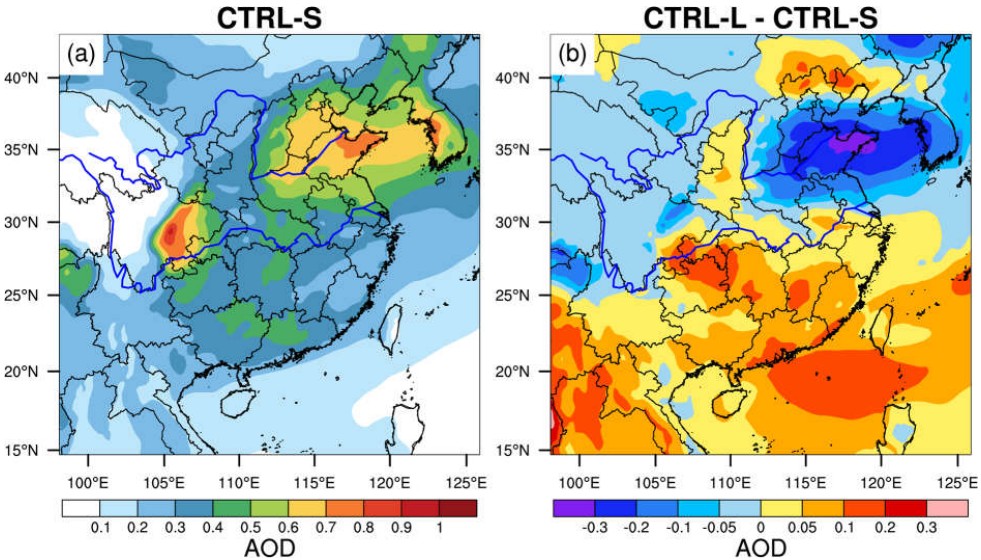


**Figure 5.** The spatial distribution of AOD for June and July of 2017 from the CTRL-S
simulation, and the difference between CTRL-L and CTRL-S.







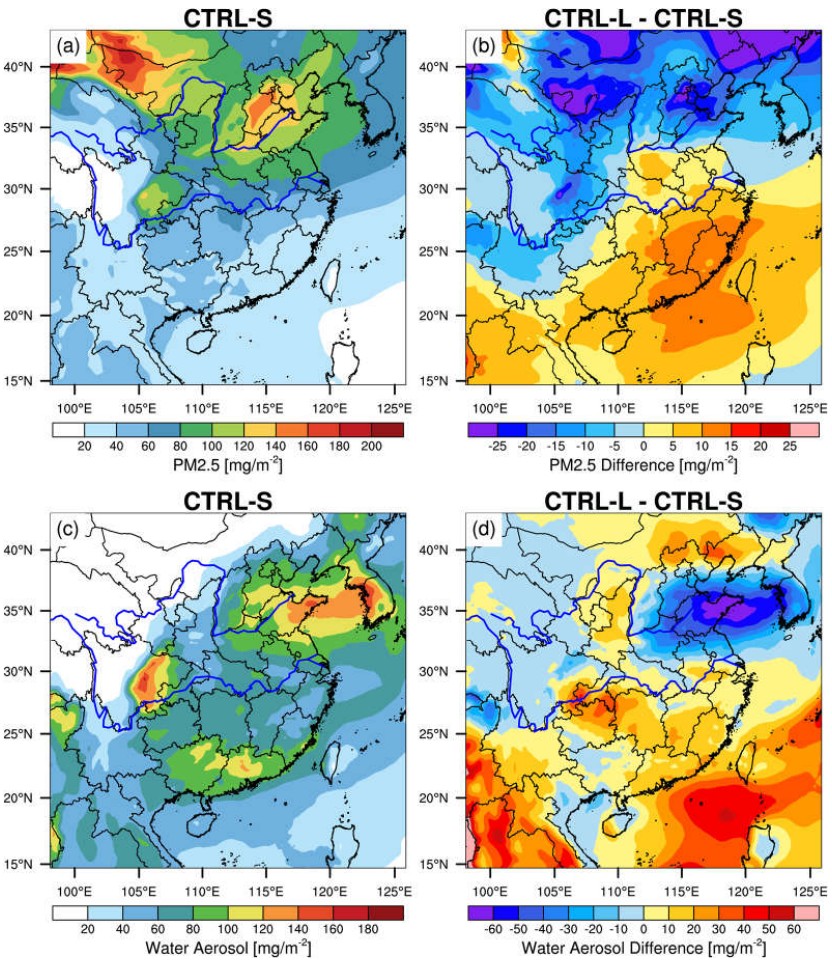

**Figure 6.** The spatial distributions of column integrated total (a) PM2.5 concentration and (c)
water content in aerosol averaged for June and July of 2017 from the CTRL-S simulation, and
(b) (d) the difference between CTRL-L and CTRL-S.




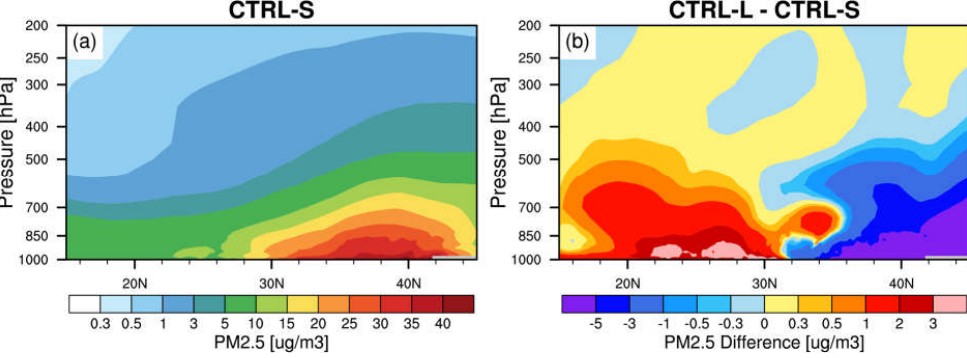

**Figure 7.** The latitude-height cross-section of (a) total PM2.5 averaged between 105°E and 122°E for June and July of 2017 from the CTRL-S experiment, and (b) the difference between CTRL-L and CTRL-S.






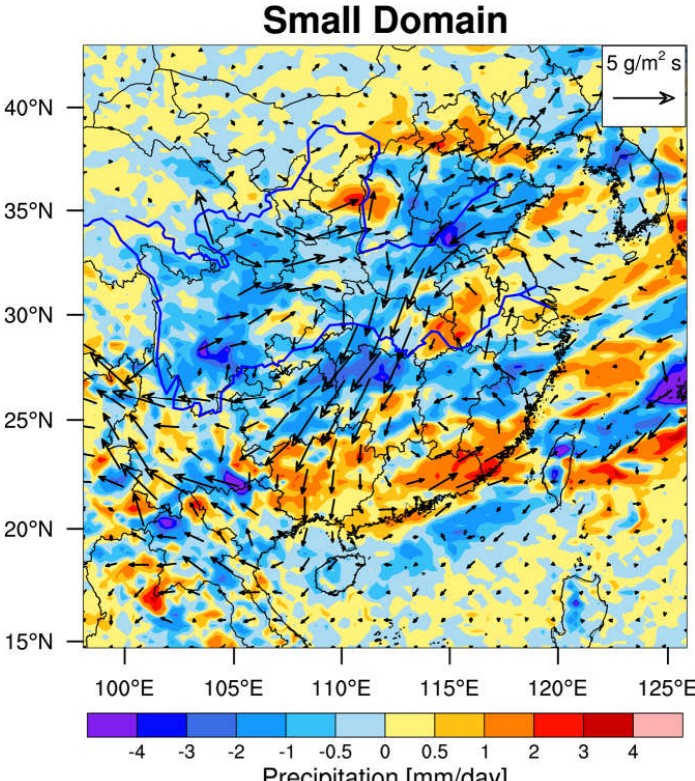

**Figure 8.** The spatial distributions of aerosol-induced difference (CTRL-CLEAN) of
precipitation and moisture transport at 700 hPa averaged for June and July of 2017 from the
small domain simulations.




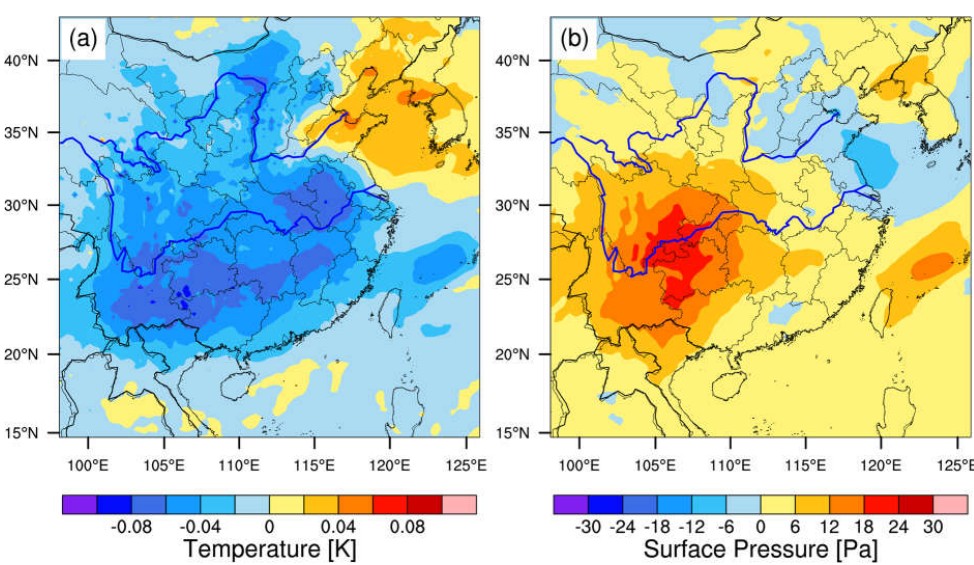

**Figure 9.** The spatial distributions of aerosol-induced difference (CTRL-CLEAN) of (a)
atmosphere temperature below 500 hPa and (b) surface pressure averaged for June and July of
2017 from the small domain simulations. We interpolate the atmosphere temperature to the
isobaric surface below 500 hPa and get the atmosphere temperature below 500 hPa by weighted
average according to the layer height.




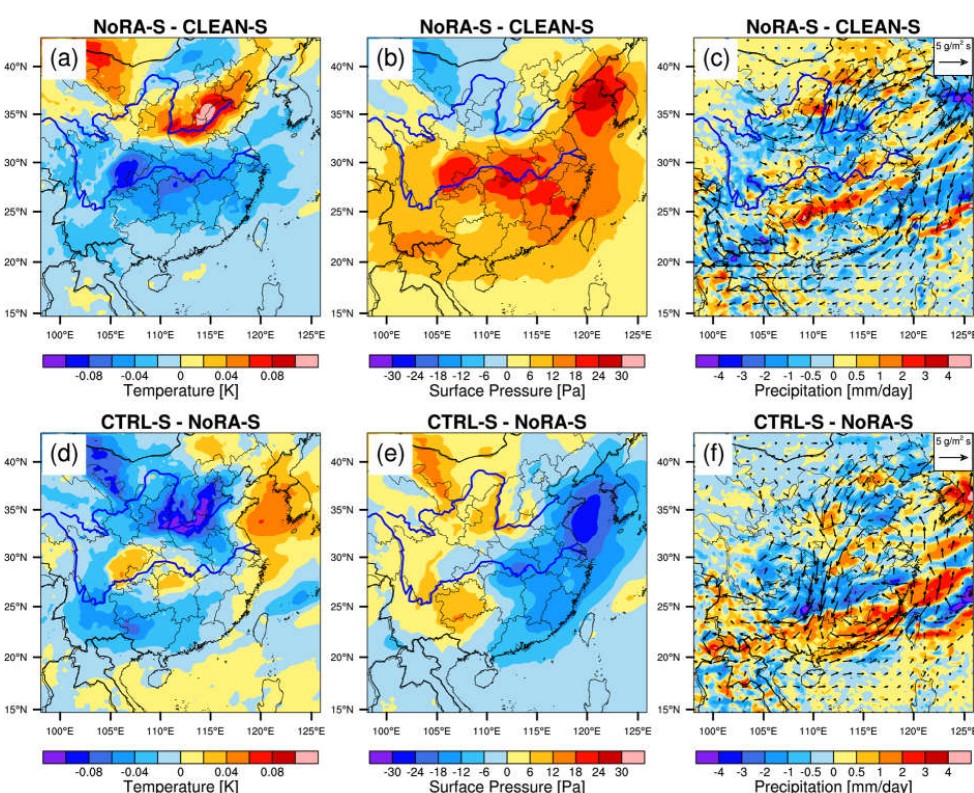

**Figure 10.** The spatial distributions of Aerosol-Cloud interactions induced difference of (a) atmosphere temperature below 500 hPa, (b) surface pressure and (c) precipitation and moisture transport at 700 hPa averaged for June and July of 2017 from the small domain simulations. And the spatial distributions of Aerosol-Radiation interactions induced difference of (d) atmosphere temperature below 500 hPa, (e) surface pressure and (f) precipitation and moisture transport at 700 hPa averaged for June and July of 2017 from the small domain simulations.




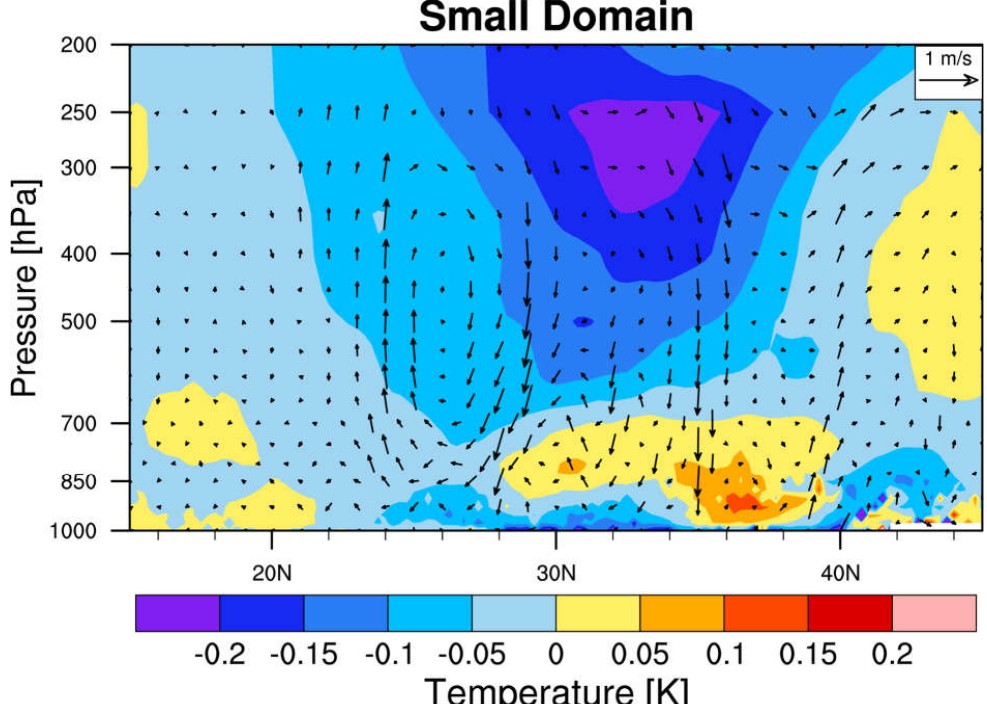

**Figure 11.** The latitude-pressure cross-section of aerosol-induced difference (CTRL-CLEAN) of temperature and wind averaged between 105°E and 122°E for June and July of 2017 from the small domain simulation.




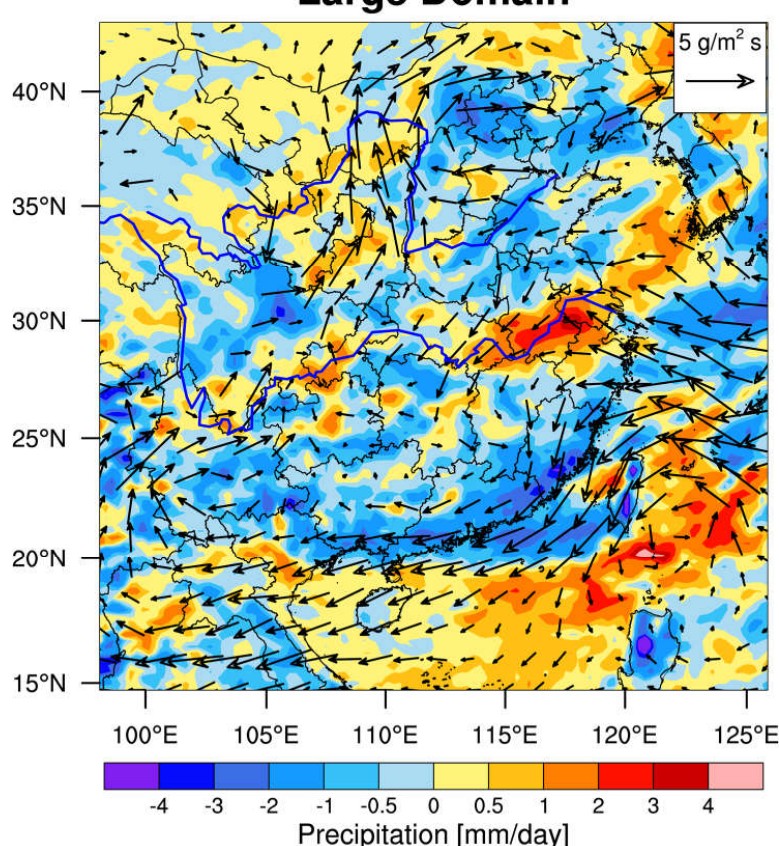

**Figure 12.** The same as figure 8, but from the large domain simulation.





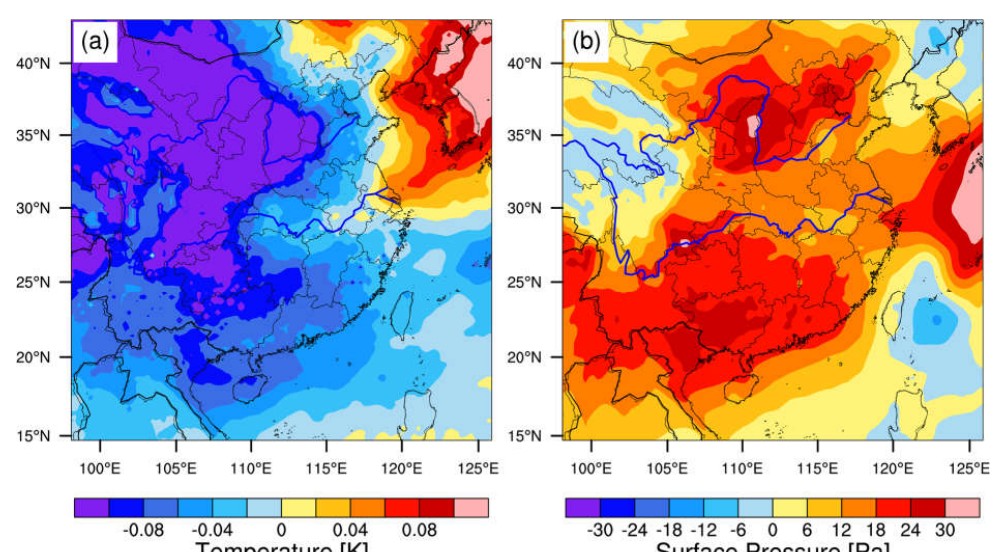

**Figure 13.** Same as Fig. 9, but from the large domain simulation.






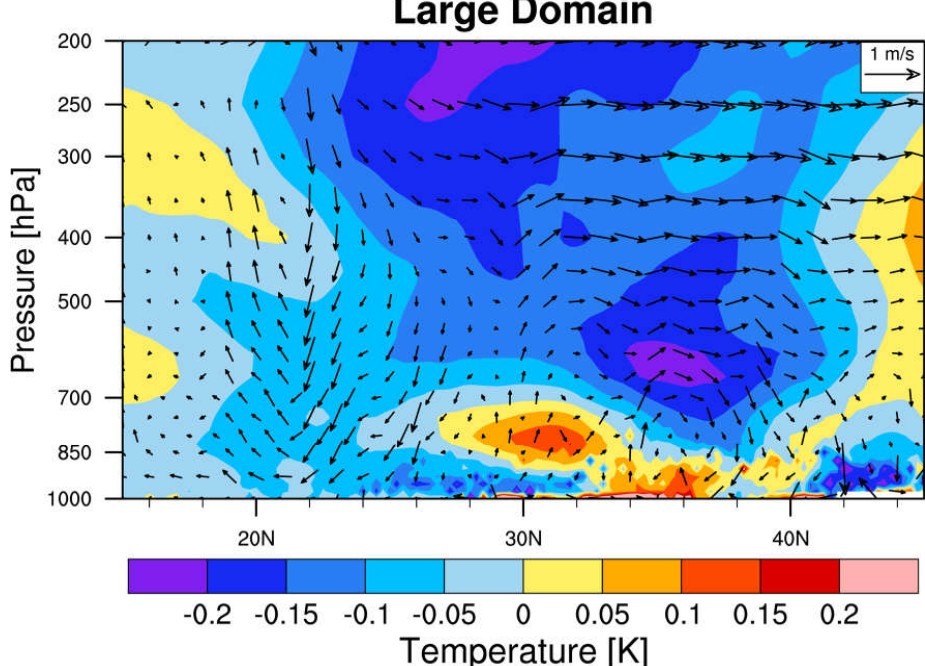

**Figure 14.** Same as figure 11, but from the large domain simulation.






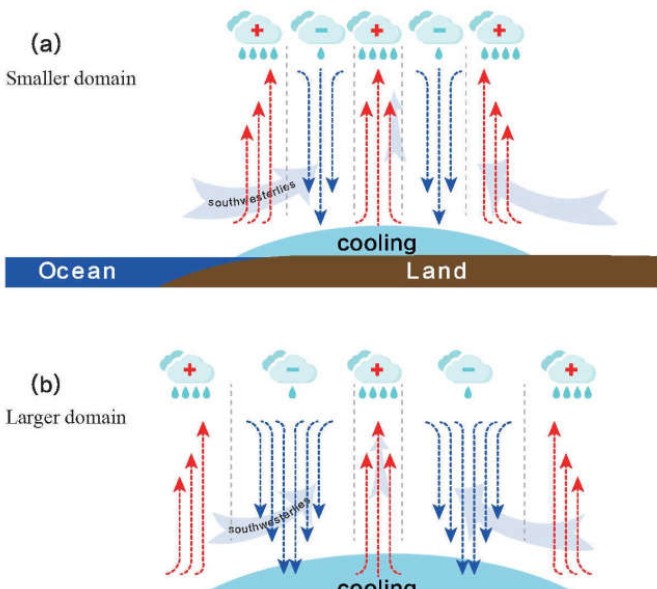

**Figure 15.** The schematic plot of aerosol impacts in (a) small domain simulation and (b) large
domain simulation over East Asia. The light blue shadow area represents the extent of aerosol
induced decrease of lower tropospheric temperature and increase of surface pressure. The red
(blue) vector dash lines represent updraft (downdraft) anomalies. The "+" ("-") above the
region indicates the aerosol-induced increase (decrease) of precipitation.