# Peer review of "Robustness of simulating aerosol climatic impacts using regional model (WRF-Chem v3.6): the sensitivity to domain size"

_Geoscientific Model Development, 2021_

## Referee Comment (RC2)

**Review comments to manuscript: gmd-2021-232**

**Robustness of simulating aerosol climatic impacts using regional model (WRF-Chem v3.6): the sensitivity to domain size**

**by Wang et al.**

**General comments:**

The manuscript seeks to investigate the impact of domain size in the regional simulations of aerosol climatic feedbacks. Specifically, the authors focus on China and seek to identify discrepancies in the simulations of precipitation spatial patterns related to the East Asian Summer monsoon (EASM) and attribute them to discrepancies in the simulations of aerosol properties. An ensemble of WRF-Chem simulations is thus generated, where the runs differ by the size of the simulated domain, by the anthropogenic emission levels and inclusion of aerosol feedbacks. The manuscript thus investigates the important and debated question of how to set up regional model simulations to properly account for aerosol impacts on meteorological variables and ultimately on the regional climate. Although the presented topic is relevant to the GMD readership, the following specific and technical comments need to be addressed to consider it suitable for publication.

**Specific comments:**

- The manuscript should be fully and carefully revised to fix the English grammar. Several sentences are either not clear, missing verbs (e.g. first sentence in the Abstract), or contain typos. Support from an English editor is needed. I will not highlight in the technical comments all the mistakes as they are too many and major rewordings are needed.
- Significant restructuring to the manuscript is needed. For example, the data used for model evaluation is not mentioned until the result section. A separate section discussing the data used should be included before/after the simulation setup. Also, more details are needed about the simulations (see technical comments below).
- The objectives of the study are not clearly stated, as initially the manuscript is presented as a pure sensitivity study, while the result section starts with a model evaluation. Do the authors want to identify which setup plays a major role in simulating different aerosol and meteorological properties or do they want to identify which setup allows for a better representation of observations? If the latter is the case, a sensitivity on the spatial resolution and/or chemistry/aerosol schemes applied needs to be also included.
- While there is prior literature evidence that boundary conditions significantly impact the spatio-temporal patterns of aerosol properties within regional model simulations, varying the domain size is only one of the possible approaches. Multiple literature studies have addressed this issue by analyzing the sensitivity to the spatial resolution applied. The authors should comment on this and expand the literature review to better characterize the ongoing research on the topic (some references are provided below).
- A spatial resolution of 30 km is applied to both the large and small domain. Is this a proper resolution to capture the spatial variability and dynamics of aerosols over the region? A discussion on why 30 km is chosen should be included. Further, simulating a larger domain implies higher computing costs, as it would occur if the author would choose a finer spatial

resolution over a smaller domain. The author should discuss the quantified biases in terms of the resources (e.g. computing cost) needed for such simulations and how the bias can be minimized based on computing costs and the domain size and resolution applied.

**Technical comments:**

- The title could be improved/reworded. The expression "robustness of simulating" is not very clear.
- Key point #2: it is not clear if a bigger or smaller domain is associated with a weaker EASM moisture transport (similarly at line 39).
- Line 41: it is not clear what is the pattern +-+-+.
- Line 76: it would be clearer to specify the time frame when these air pollution episodes have occurred.
- Line 84: what do you mean by "extraterrestrial natural forcing"?
- Line 113: This paragraph should be revised. What is the impact of the much more simplified aerosol representation in GCMs? Aerosols scales of variability are generally not reproduced by GCMs, so regional simulations may be expected to perform better.
- Line 130: add "lateral boundary conditions".
- Line 131: other literature studies addressing the issue of spatial resolution and parameterizations applied are:
  - Di Luca, A., de Elía, R., and Laprise, R.: Challenges in the Quest for Added Value of Regional Climate Dynamical Downscaling, Curr. Clim. Change Rep., 1, 10–21, doi:10.1007/s40641-015- 0003-9, 2015.
  - Crippa, P., Sullivan, R. C., Thota, A., and Pryor, S. C.: The impact of resolution on meteorological, chemical and aerosol properties in regional simulations with WRF-Chem, Atmos. Chem. Phys., 17, 1511–1528, https://doi.org/10.5194/acp-17-1511-2017, 2017.
  - Diaconescu, E. and Laprise, R.: Can added value be expected in RCM-simulated large scales?, Clim. Dynam., 41, 1769–1800, doi:10.1007/s00382-012-1649-9, 2013.
  - Crippa, P., Sullivan, R. C., Thota, A., & Pryor, S. C. (2019). Sensitivity of simulated aerosol properties over eastern North America to WRF-Chem parameterizations. Journal of Geophysical Research: Atmospheres, 124, 3365– 3383. https://doi.org/10.1029/2018JD029900

- Line 142: which horizontal and vertical resolution did that study apply?
- Line 202: there is no mention of the applied resolution in the description of the simulation setup. Also, the authors do not specify the chemical scheme applied which is also important for the simulation of aerosol properties. Finally, in the result section there is mention of runs performed without aerosol feedback on, so those simulations should be included when presenting the ensemble.
- Line 219: it is not clear how initial conditions are changed. Is the 12-16 May the spin up time? How do you initialize the runs on June 1st? This sentence needs to be rephrased and clarified.

- Line 257: observations should be introduced in the data and methods part and the manuscript's objectives should be revised to include the model evaluation component.
- Line 273: is the underestimation due to the role of aerosols and the fact that lower emissions are provided as input to the model? How do observations compare to the simulations with "real" emissions? It is not clear why the CLEAN scenarios are used for model evaluation instead of the "real" ones.
- PM2.5 needs a subscript through the all manuscript.
- Section 3.2. There was no mention that $PM_{2.5}$ and AOD would be used as metrics for model evaluation, so this idea should be anticipated earlier in the text. Also, it is not clear why the authors compare $PM_{2.5}$ against AOD instead of performing a proper evaluation against observations from the ground.
- Line 371: the NoRA experiments were never introduced before, but they are part of the ensemble setup.
- Line 416-421: what is the role of aerosol composition on the radiative impacts?
- Line 430: the pattern +-+-+ is not clear

---

## Author Comment (AC1)

General comments:

• This study investigates the impacts of domain size on the modeling of aerosol impacts on East Asian summer monsoon using the updated regional model WRF-Chem. The authors compare the simulation results with small and large domain size. The consistent side in different domain size simulations is that aerosols lead to the cooling of lower troposphere and thus the anti-cyclone circulation anomalies and the weakening of EASM moisture transport. The results also demonstrate that domain size has a great influence on the simulated meteorological fields which leads to the difference of simulated strength and area extent of aerosol-induced changes of lower-tropospheric temperature and pressure, which further results in different locations of circulation and precipitation anomalies. This study gives a highlight to understand the importance of domain size and proper modeling of meteorological fields in the study of aerosol impacts on circulation and precipitation and has a good guiding significance for similar research. There are a few questions needed to be revised.

We thank the reviewer for the detailed review. The comments are very helpful to improve the quality of the manuscript. Now we revise the manuscript upon all the specific comments listed below.

**Specific comments:**

• In the description of figure 3 and 4, there are some information about the results from CLEAN-L, such as that at Line 303-305. I suggest the author to revise figure 3 and 4 by adding the results from ERA5 and CLEAN-L, and then figure S2 and S3 can be removed.

Thanks for your great suggestion. Fig. 3 and Fig. 4 are revised now to include the ERA5 results. Fig. S2 and S3 are deleted in the revised support file. For the results from the simulation with larger domain size (i.e., CLEAN-L), we keep showing the difference between CLEAN-L and CLEAN-S, because the difference between the two experiments is the focus of this study.

**• In figure 4 and S3, are the temperature and wind averaged for June and July? Please clarify it.**

In Fig. 4 and S3, the temperature and wind are averaged for June and July. Now we clarify it in the text and captions.

• At line 348-349, the author claimed that "At 32°N-36•N, CTRL-L simulates lower aerosol mass concentration near the surface and higher above 850 hPa.". Is there any explanation for this?

Thanks for the checking. There is a typo here. Now we clarify it and add some explanation

in the text as "At 32°N-36°N, CTRL-L simulates lower aerosol mass concentration near the surface and higher between 700 hPa and 850 hPa, likely due to the difference in aerosol wet scavenging and transport between the two experiments."

• At line 382-384, I think the description "aerosol-cloud interaction also increases cloud amounts over Northeast China and its adjacent ocean (Fig. S6a) and thus reduces the lower tropospheric temperature and increases the surface pressure over the area." is not accuracy. The cloud amount increases not only over Northeast China and its adjacent ocean but also the north of Hebei province while the temperature increases over part of Northeast China and Bohai sea.

Thanks for your comment. In the text, we have the statement "The surface pressure over the Yellow River Basin is reduced slightly by aerosol-cloud interaction due to the reduction of cloud amounts (Fig. S4a) and the increase of lower tropospheric temperature". We think that the increase of temperature over part of Northeast China and Bohai sea is due to the reduction of cloud amounts over the Yellow River Basin.

Overall, our experiments generally demonstrate that aerosol-cloud interaction leads to the increase of cloud amounts and thus results in the cooling of lower tropospheric temperature. However, this does not mean that the co-locations of adjustment of cloud amounts and temperature would be simply straightforward. In a fully coupled system, the change of cloud formation due to aerosols would also adjust the temperature through the release of latent heat in the atmosphere. In addition, the change of temperature would also modulate the circulation and further feedback to the distributions of cloud and temperature.

Now we add this explanation in the revised manuscript "Although, the experiments can generally demonstrate that aerosol-cloud interaction can largely affect cloud amount, lower-tropospheric temperature, and surface pressure, please note that the co-locations of the changes of cloud, temperature, and surface pressure may not be simply straightforward. For example, in a fully coupled system, the cloud change due to aerosols would also adjust the temperature through the release of latent heat in the atmosphere. In addition, the change of temperature would also modulate the circulation and further feedback to the distributions of cloud and temperature."

**• At Line 384-386, what is the reason for the aerosol-cloud interaction-induced the reduction of cloud amounts and the increase of lower tropospheric temperature over the Yellow River Basin?**

As our response to the comments above, overall, our experiments generally demonstrate that aerosol-cloud interaction leads to the increase of cloud amounts. However, this does not mean that we can simply co-locate all the adjustment of cloud amounts and aerosol-cloud interactions. The aerosol-cloud interaction adjusts the cloud amounts and temperature, which would also modulate the circulation and further feedback to the distributions of cloud and temperature. This may lead to the reduction of cloud amounts

and the increase of lower-tropospheric temperature over the Yellow River Basin. We add more explanation in the revised manuscript as the response above.

• Please check the caption of Figure 4. I think it should be "(a)The cross-section of meridional temperature anomalies and wind averaged for 105°E and 122°E from the CLEAN-S simulation, and (b) the difference of temperature (not meridional temperature anomalies) and wind between CLEAN-L and CLEAN-S. The meridional temperature anomalies are calculated by subtracting the mean temperature in this latitude range at each pressure level."

Thanks a lot for your correction. Now the caption is revised as "(a, b) The cross-section of meridional temperature anomalies and wind averaged for 105°E and 122°E from the ERA5 reanalysis and the CLEAN-S simulation during June to July, and (c) the difference of temperature (not meridional temperature anomalies) between CLEAN-L and CLEAN-S. The meridional temperature anomalies are calculated by subtracting the mean temperature in this latitude range at each pressure level."

• At Line 376, "the spatial distribution the impacts of aerosol-cloud" should be "the spatial distribution of the impacts of aerosol-cloud".

Thanks for your correction. Now the text is revised as "the spatial distributions of the impact of aerosol-cloud and aerosol-radiation interactions on …".

**Anonymous Referee #2**

General comments:

• The manuscript seeks to investigate the impact of domain size in the regional simulations of aerosol climatic feedbacks. Specifically, the authors focus on China and seek to identify discrepancies in the simulations of precipitation spatial patterns related to the East Asian Summer monsoon (EASM) and attribute them to discrepancies in the simulations of aerosol properties. An ensemble of WRF-Chem simulations is thus generated, where the runs differ by the size of the simulated domain, by the anthropogenic emission levels and inclusion of aerosol feedbacks. The manuscript thus investigates the important and debated question of how to set up regional model simulations to properly account for aerosol impacts on meteorological variables and ultimately on the regional climate. Although the presented topic is relevant to the GMD readership, the following specific and technical comments need to be addressed to consider it suitable for publication.

We thank the reviewer for the detailed review. The comments are very helpful to improve the quality of the manuscript. Now we revise the manuscript upon all the specific comments listed below. A new section about the observations and reanalysis used and more clarifications are added in the revised manuscript.

**Specific comments:**

• The manuscript should be fully and carefully revised to fix the English grammar. Several sentences are either not clear, missing verbs (e.g. first sentence in the Abstract), or contain typos. Support from an English editor is needed. I will not highlight in the technical comments all the mistakes as they are too many and major rewordings are needed.

Thank the reviewer for the detailed checking of written English. We have carefully revised the manuscript to fix the problems of grammar.

• Significant restructuring to the manuscript is needed. For example, the data used for model evaluation is not mentioned until the result section. A separate section discussing the data used should be included before/after the simulation setup. Also, more details are needed about the simulations (see technical comments below).

Thanks for your suggestion. A new section "2.4 Observations and reanalysis" is added in the revised manuscript to describe the datasets used. More details about the simulations and other clarification are also added as the response to the specific comments below.

• The objectives of the study are not clearly stated, as initially the manuscript is presented as a pure sensitivity study, while the result section starts with a model evaluation. Do the authors want to identify which setup plays a major role in simulating different aerosol and meteorological properties or do they want to

**identify which setup allows for a better representation of observations? If the latter is the case, a sensitivity on the spatial resolution and/or chemistry/aerosol schemes applied needs to be also included.**

Sorry for the confusion, and thanks for the suggestion. As we stated in the introduction and also in the title, the aim of this study is to examine the sensitivities of regional modeling results of aerosol impact to regional domain size and is not going to evaluate the simulation results to determine the optimal model configuration for the experiments. However, some observations and reanalysis datasets are still used to provide the references for the key fields. The comparison with these references can demonstrate whether the simulation results are acceptable for further analysis. This is why we include the observations and reanalysis in the results presented (Fig. 2-4). Now we add more clarification in the revised manuscript as

"Although the aims of this study are not evaluating the simulation results to determine the optimal model configuration for the experiments, some observations and reanalysis datasets are still used to provide the references for the key fields. The comparison with these references can demonstrate whether the simulation results are acceptable for further analysis."

"More specifically, on one hand, larger-domain simulation may better allow feedbacks of aerosol impact on weather and climate systems without strong lateral boundary constraint (e.g., Seth and Giorgi, 1998; Leduc and Laprise, 2009; Diaconescu et al., 2013), but it may produce biased meteorological fields compared to smaller-domain simulation, which can then significantly influence the modeling results of aerosol impact. On the other hand, although the simulation with smaller domain produces better large-scale circulation compared to the reanalysis, the lateral boundary may also have stronger constraint on aerosol feedbacks to large-scale circulation. Therefore, not like meteorological fields or aerosol properties, there is no direct observation or reanalysis that can be as the references to evaluate aerosol impact and the optimal configuration of simulation domain is hard to be determined in this study. (Di Luca et al., 2015; Crippa et al., 2017)."

• While there is prior literature evidence that boundary conditions significantly impact the spatio-temporal patterns of aerosol properties within regional model simulations, varying the domain size is only one of the possible approaches. Multiple literature studies have addressed this issue by analyzing the sensitivity to the spatial resolution applied. The authors should comment on this and expand the literature review to better characterize the ongoing research on the topic (some references are provided below).

Thanks a lot for providing these very interesting and informative references. Although there have been many studies examining the impact of model configuration on simulation results, the focus of this study is on the impact of simulation domain size. Even though, quite a few studies investigated the impact of domain size on simulated meteorological fields and aerosol properties, its impact on aerosol effect is first time studied. Therefore, we don't plan to spend too much effort on reviewing the literatures about all the influential factors in regional modeling.

We have went through all of these papers provided by the reviewer and cited them in the revised manuscript. More discussion is also added in multiple places of the revised manuscript, for example, "Crippa et al. (2017) found that enhanced resolution (from 60 to 12 km) can improve the model performance of meteorological fields and aerosol optical depth (AOD).", and "Although the modeling results of aerosol impact in this study may have some uncertainties associated with physical and chemical processes, emissions, and horizontal resolutions (e.g., Di Luca et al., 2015; Crippa et al., 2019), it highlights the impact of simulation domain size on regional modeling aerosol impact on monsoonal circulation and precipitation, ..."

• A spatial resolution of 30 km is applied to both the large and small domain. Is this a proper resolution to capture the spatial variability and dynamics of aerosols over the region? A discussion on why 30 km is chosen should be included. Further, simulating a larger domain implies higher computing costs, as it would occur if the author would choose a finer spatial resolution over a smaller domain. The author should discuss the quantified biases in terms of the resources (e.g. computing cost) needed for such simulations and how the bias can be minimized based on computing costs and the domain size and resolution applied.

Thanks to raise this point. Horizontal resolution is definitely an important factor that may affect the simulation results. However, the focus of this study is on domain size and therefore, we do not spend much efforts to investigate the impact of resolution. For the domain sizes used in this study, particularly for the larger one, 30 km is a reasonable choice considering the balance of computational efficiency and modeling performance. The comparable horizontal resolutions have also been widely used for investigating aerosol impact on regional climate (e.g., Zhang et al., 2009; Stanelle et al., 2010; Zhao et al., 2011, 2012; Chen et al., 2014; Wang et al., 2015).

We acknowledged the impact from other factors including horizontal resolution in discussion part as "Although the modeling results of aerosol impact in this study may have some uncertainties associated with physical and chemical processes, emissions, and horizontal resolutions (e.g., Di Luca et al., 2015; Crippa et al., 2019), it highlights the impact of simulation domain size on regional modeling aerosol impact on monsoonal circulation and precipitation, which may not be limited to the region of East Asia."

Now we add the clarification of the selection of 30 km horizontal resolution in the Section 2.2 as "The horizontal resolution of 30 km is selected for both simulation domains with the consideration of the balance of computational efficiency and modeling performance, particularly for the larger domain. The comparable horizontal resolutions have also been widely used for investigating aerosol impact on regional climate (e.g., Zhang et al., 2009;

**Stanelle et al., 2010; Zhao et al., 2011, 2012; Chen et al., 2014; Wang et al., 2015)."**

In terms of designing two sets of experiments with the same horizontal resolution but different domain sizes, we agree that the larger domain size will cost more computational resource. However, in fact, the aim of this study is different, to some extent, from previous studies that also investigated the impact of domain size. Previous studies often evaluated the simulation results using different domain sizes with observations and reanalysis to determine an optimal configuration or estimate the added values from regional simulation. However, the optimal configuration of simulation domain for modeling aerosol impact is relatively harder to be determined, because, not like meteorological fields or aerosol properties, there is no direct observation or reanalysis that can be as the references to evaluate aerosol impact. The intention of increasing the smaller domain size to the larger one is to release the strong constraint from the lateral boundary and thus expect a better representation of aerosol feedbacks to large-scale circulation. Although the results show that the simulation with smaller domain produces better large-scale circulation compared to the reanalysis, the lateral boundary may also have stronger constraint on and thus limit the aerosol feedbacks to large-scale circulation. Therefore, we state that it needs the effort to improve the simulated meteorological fields with larger regional domain or global domain in order to model robust aerosol climatic impact.

Now we revise the discussion about this in the text as "More specifically, on one hand, larger-domain simulation may better allow feedbacks of aerosol impact on weather and climate systems without strong lateral boundary constraint (e.g., Seth and Giorgi, 1998; Leduc and Laprise, 2009; Diaconescu et al., 2013), but it may produce biased meteorological fields compared to smaller-domain simulation, which can then significantly influence the modeling results of aerosol impact. On the other hand, although the simulation with smaller domain produces better large-scale circulation compared to the reanalysis, the lateral boundary may also have stronger constraint on aerosol feedbacks to large-scale circulation. Therefore, not like meteorological fields or aerosol impact (Di Luca et al., 2015; Crippa et al., 2017), and the optimal configuration of simulation domain is hard to be determined in this study. It may be the key to improve the simulated meteorological fields with larger regional domain or global domain in order to model robust aerosol climatic impact."

**• The title could be improved/reworded. The expression "robustness of simulating" is not very clear.**

Thanks for the suggestion. The title is changed to "The sensitivity of simulated aerosol climatic impact to domain size using regional model (WRF-Chem v3.6)".

• Key point #2: it is not clear if a bigger or smaller domain is associated with a weaker

**EASM moisture transport (similarly at line 39).**

As we stated, the simulations with both smaller and larger domains demonstrate consistently that aerosols weaken EASM moisture transport, which is highlighted by the key point. Due to the limit words of key point, we cannot put all the findings in it. In terms of the impact of domain size on EASM monsoon, it is stated in the abstract "For example, the simulation with increasing domain size produces weaker EASM circulation, which also affect aerosol distributions significantly."

**• Line 41: it is not clear what is the pattern +-+-+.**

Now, it is revised as "The aerosol-induced adjustment of monsoonal circulation results in an alternate increase and decrease pattern of precipitation over the continent of China."

• Line 76: it would be clearer to specify the time frame when these air pollution episodes have occurred.

Now, it is clarified as "Due to the large population and the rapid economic development in last few decades, ...".

**• Line 84: what do you mean by "extraterrestrial natural forcing"?**

It means the natural forcing from the outside of the Earth, for example, the change of solar radiation.

• Line 113: This paragraph should be revised. What is the impact of the much more simplified aerosol representation in GCMs? Aerosols scales of variability are generally not reproduced by GCMs, so regional simulations may be expected to perform better.

We agree that the difference between GCMs and regional models can be resulted from many factors, such as different representations of aerosols and also physical processes. However, one key reason of using regional models instead of GCMs may be their relatively higher horizontal resolution. Therefore, we highlight this point here. In this paragraph, we don't plan to review all the difference between GCMs and regional models.

**• Line 130: add "lateral boundary conditions".**

Revised.

• Line 131: other literature studies addressing the issue of spatial resolution and parameterizations applied are:

• Di Luca, A., de Elía, R., and Laprise, R.: Challenges in the Quest for Added Value of Regional Climate Dynamical Downscaling, Curr. Clim. Change Rep., 1, 10–21, doi:10.1007/s40641-015-0003-9, 2015.

• Crippa, P., Sullivan, R. C., Thota, A., and Pryor, S. C.: The impact of resolution

on meteorological, chemical and aerosol properties in regional simulations with WRF-Chem, Atmos. Chem. Phys., 17, 1511–1528, https://doi.org/10.5194/acp-17-1511-2017, 2017.

Diaconescu, E. and Laprise, R.: Can added value be expected in RCM-simulated large scales?, Clim. Dynam., 41, 1769–1800, doi:10.1007/s00382-012-1649-9,2013.
Crippa, P., Sullivan, R. C., Thota, A., & Pryor, S. C. (2019). Sensitivity of simulated aerosol properties over eastern North America to WRF-Chem parameterizations. Journal of Geophysical Research: Atmospheres, 124, 3365–3383. https://doi.org/10.1029/2018JD029900

Thanks a lot for providing these very interesting and informative references. We have went through all of these papers and cited them in the revised manuscript. More discussion is also added in the revised manuscript as "More specifically, on one hand, larger-domain simulation may better allow feedbacks of aerosol impact on weather and climate systems without strong lateral boundary constraint (e.g., Seth and Giorgi, 1998; Leduc and Laprise, 2009; Diaconescu et al., 2013), but it may produce biased meteorological fields compared to smaller-domain simulation, which can then significantly influence the modeling results of aerosol impact. On the other hand, although the simulation with smaller domain produces better large-scale circulation compared to the reanalysis, the lateral boundary may also have stronger constraint on aerosol feedbacks to large-scale circulation. Therefore, not like meteorological fields or aerosol properties, there is no direct observation or reanalysis that can be as the references to evaluate aerosol impact (Di Luca et al., 2015; Crippa et al., 2017), and the optimal configuration of simulation domain is hard to be determined in this study."

**• - Line 142: which horizontal and vertical resolution did that study apply?**

In Bhaskaran et al. (2012), their model resolutions are 60 km for the larger domain experiment and 50 km for the smaller domain experiment, respectively. They set 14 vertical layers in the experiments.

Line 202: there is no mention of the applied resolution in the description of the simulation setup. Also, the authors do not specify the chemical scheme applied which is also important for the simulation of aerosol properties. Finally, in the result section there is mention of runs performed without aerosol feedback on, so those simulations should be included when presenting the ensemble.

Thanks for your suggestion. The details of simulation setup can be found in Table 2 as we state in the text "The simulation configuration is summarized in Table 2", including horizontal resolution and chemical scheme. Now, the horizontal resolution is explicitly mentioned in the text, and the reason of this selection is also discussed in the revised manuscript as our response to other comments. The chemical scheme used in the study is also stated in Section 2.1. The description of the experiment without aerosol-radiation

interaction is also added in the revised manuscript as "Besides these experiments, another set of experiment NoRA-S is conducted to isolate aerosol-radiation and aerosol-cloud interactions for further understanding the mechanisms of aerosol impact, which is also listed in Table 1."

**• - Line 219: it is not clear how initial conditions are changed. Is the 12-16 May the spin up time? How do you initialize the runs on June 1 st? This sentence needs to be rephrased and clarified.**

Sorry for the confusion. They are all continuous simulations, but starts at different days. The simulation results for June and July are analyzed. Now it is clarified in the revised manuscript as "Five ensemble simulations are performed for each experiment by changing the simulation initial time at UTC 0000 from May 12th to May 16th, 2017 (i.e., the five ensemble simulations start at UTC 0000 of May 12th, 13th, 14th, 15th, and 16th, respectively). The averaged results from five ensembles are analyzed to reduce the influence of modeling internal variability. The simulations run continuously through entire June and July of 2017."

**• - Line 257: observations should be introduced in the data and methods part and the manuscript's objectives should be revised to include the model evaluation component.**

Thanks for the suggestion. As we stated in the introduction and also in the title, the aim of this study is to examine the sensitivities of regional modeling results of aerosol impact to regional domain size and is not going to evaluate the simulation results to determine the optimal model configuration for the experiments. However, some observations and reanalysis datasets are still used to provide the references for the key fields. The comparison with these references can demonstrate whether the simulation results are acceptable for further analysis. Please also note that the optimal configuration of simulation domain for modeling aerosol impact is relatively harder to be determined, because, not like meteorological fields or aerosol properties, there is no direct observation or reanalysis that can be as the references to evaluate aerosol impact.

Now we add Section 2.4 about observations and reanalysis and also more clarification in the revised manuscript as

"Although the aims of this study are not evaluating the simulation results to determine the optimal model configuration for the experiments, some observations and reanalysis datasets are still used to provide the references for the key fields. The comparison with these references can demonstrate whether the simulation results are acceptable for further analysis. The MISR (Multi-angle Imaging SpectroRadiometer, instrument on board the NASA Terra platform) retrieval dataset is used as a reference of spatial distribution of AOD (Diner et al, 1998; Martonchik et al., 2004). When showing the comparison between the MISR retrieved and the simulated AOD, the simulation results are sampled from 10 am - 11 am for averaging and at the locations of the retrievals because the Terra platform passes

over the equator at about 10:45 LT (Diner et al, 2001). The precipitation datasets of CMA (National Meteorological Information Center of China Meteorological Administration) and CMORPH (Climate Prediction Center MORPHing technique) are used as the references for spatial and temporal variations of precipitation during the simulation period. The CMORPH dataset is a global precipitation reanalysis dataset that is derived from geostationary satellite IR imagery (Joyce et al., 2004). The CMA rainfall was measured by tipping buckets, self-recording siphon rain gauges, or automatic rain gauges and was subject to strict quality control. The European Centre for Medium-Range Weather Forecasts (ECMWF) Reanalysis (ERA5) are used as a reference of wind fields (Hersbach, 2020)."

"More specifically, on one hand, larger-domain simulation may better allow feedbacks of aerosol impact on weather and climate systems without strong lateral boundary constraint (e.g., Seth and Giorgi, 1998; Leduc and Laprise, 2009; Diaconescu et al., 2013), but it may produce biased meteorological fields compared to smaller-domain simulation, which can then significantly influence the modeling results of aerosol impact. On the other hand, although the simulation with smaller domain produces better large-scale circulation compared to the reanalysis, the lateral boundary may also have stronger constraint on aerosol feedbacks to large-scale circulation. Therefore, not like meteorological fields or aerosol properties, there is no direct observation or reanalysis that can be as the references to evaluate aerosol impact (Di Luca et al., 2015; Crippa et al., 2017), and the optimal configuration of simulation domain is hard to be determined in this study."

• Line 273: is the underestimation due to the role of aerosols and the fact that lower emissions are provided as input to the model? How do observations compare to the simulations with "real" emissions? It is not clear why the CLEAN scenarios are used for model evaluation instead of the "real" ones.

Both CLEAN-S and CLEAN-L experiments are without aerosol impact, i.e., emissions of aerosol and its precursors are reduced to the 10% of original. Therefore, the underestimation in CLEAN-L is not due to aerosol impact.

We would like to emphasize again that, as our response to the comments above, the aims of this study are not evaluating the simulation results to determine the optimal model configuration for the experiments. Some observations and reanalysis datasets are used in order to provide the references for the key fields, so that the comparison with these references can demonstrate whether the simulation results are acceptable for further analysis. Please note that the optimal configuration of simulation domain for modeling aerosol impact is relatively harder to be determined, because, not like meteorological fields or aerosol properties, there is no direct observation or reanalysis that can be as the references to evaluate aerosol impact. In addition, the modeling biases from many other factors, as the reviewer also raised in other comments, can cancel out each other, so that the indirect observations (e.g., precipitation) are not appropriate to be used for evaluating aerosol impact. For example, if the simulation with aerosol impact and smaller domain produces better precipitation compared to the one with larger domain, we cannot say the aerosol impact simulated by the smaller domain is more reasonable, because it may be due to the positive bias from the smaller domain cancels out the negative bias from the aerosol impact. Therefore, in this study, we start from the comparison between the two experiments without aerosol impact, because this reflects the impact of domain size on the meteorological fields without considering aerosol impact. And then, we move to the analysis of influence of domain size on meteorological fields with aerosol impact. As we stated in the introduction and also in the title, the aim of this study is to examine the sensitivities of regional modeling results of aerosol impact to regional domain size, instead of evaluating the modeling results. We also add more clarification in the revised manuscript as our response to other comments of the reviewer.

**• - PM2.5 needs a subscript through the all manuscript.**

Thanks for your checking. The full text has been revised

• - Section 3.2. There was no mention that PM 2.5 and AOD would be used as metrics for model evaluation, so this idea should be anticipated earlier in the text. Also, it is not clear why the authors compare PM 2.5 against AOD instead of performing a proper evaluation against observations from the ground.

As we respond above, the aims of this study are not evaluating the simulation results to determine the optimal model configuration for the experiments. Some observations and reanalysis datasets are used only in order to provide the references for the key fields.

AOD, instead of ground  $PM_{2.5}$ , is selected as a metric because it is directly associated with aerosol-radiation interaction. The integrated column mass of  $PM_{2.5}$  and associated water aerosol mass are analyzed between the experiments because they can help understand the difference in AOD between the experiments.

**• - Line 371: the NoRA experiments were never introduced before, but they are part of the ensemble setup.**

Thanks for your suggestion. Now, the description of the experiment without aerosolradiation interaction is added in the revised manuscript as "Besides these experiments, another set of experiment NoRA-S is conducted to isolate aerosol-radiation and aerosolcloud interactions for further understanding the mechanisms of aerosol impact, which is also listed in Table 1."

**• - Line 416-421: what is the role of aerosol composition on the radiative impacts?**

This is a great comment. The different aerosol components would definitely play different roles in aerosol-radiation interactions, which is an interesting topic and has been explored by some previous studies with global and regional models. However, the investigation of

this topic is beyond the scope of this study. We may explore this in future.

**• - Line 430: the pattern +-+-+ is not clear**

Now, it is revised as "Over the continent of China, aerosol induces an alternate increase and decrease pattern (denoted as "+-+-+") of precipitation changes, i.e., precipitation increases in the south of 25°N, north of 40°N, and around 30°N, while decreases at  $25^{\circ}N\sim30^{\circ}N$  and  $32^{\circ}N\sim40^{\circ}N$ ."